# *Perilla frutescens* Seed Residue Extract Restores Gut Microbial Balance and Enhances Insulin Function in High-Fat Diet and Streptozotocin-Induced Diabetic Rats

**DOI:** 10.3390/ijms26178176

**Published:** 2025-08-22

**Authors:** Pattharaphong Deethai, Chatsiri Siriwathanakul, Pornsiri Pitchakarn, Arisa Imsumran, Ariyaphong Wongnoppavich, Sivamoke Dissook, Teera Chewonarin

**Affiliations:** Department of Biochemistry, Faculty of Medicine, Chiang Mai University, Chiang Mai 50200, Thailand; pattharaphong_d@cmu.ac.th (P.D.); chatsiri_s@cmu.ac.th (C.S.); pornsiri.p@cmu.ac.th (P.P.); arisa.bonness@cmu.ac.th (A.I.); ariyaphong.w@cmu.ac.th (A.W.); sivamoke.dis@cmu.ac.th (S.D.)

**Keywords:** *Perilla frutescens* seed residue, diabetes, anti-inflammation, gut microbiome stability

## Abstract

The seed residue of *Perilla frutescens* possesses diverse biological properties and is rich in bioactive phytochemicals, including luteolin, rosmarinic acid, and apigenin. The aim of this study was to investigate the anti-diabetic effects of perilla seed residue crude extract (PCE) and its impact on the composition of the gut microbiome in rats with diabetes induced by a high-fat diet (HFD) and streptozotocin (STZ). Forty male Wistar rats were fed on an HFD for six weeks before receiving an injection of STZ injection to induce diabetes. These rats were then treated for four weeks with metformin (100 mg/kg bw) or PCE (100 and 1000 mg/kg bw) alongside a control group maintained on a normal diet. The results showed that PCE treatment improved metabolic parameters in diabetic rats, as evidenced by reduced water and food intake, increased body weight gain, lower blood glucose levels, and enhanced insulin secretion effects, especially at the 100 mg/kg bw dosage. PCE also promoted the regeneration of pancreatic β-cells and improved utilization of glucose. PCE also suppressed inflammation and oxidative stress, enhanced antioxidant capacity, and reduced circulating triglyceride levels. Most notably, PCE administration increased gut microbial diversity and shifted the microbiome closer to that of healthy controls, demonstrating its prebiotic effect. It promoted the abundance of beneficial bacteria that are linked to improved glucose metabolism and reduced inflammation—specifically, *Bacteroides fragilis*, *Lactobacillus*, *Clostridium*, and *Akkermansia.* Harmful bacteria associated with inflammation and poor glycemic control were reduced. Collectively, these results suggest that PCE not only helps restore a balanced gut microbiome but also offers metabolic benefits that could improve diabetic outcomes. These findings position PCE as a promising supplement for the management of diabetes and encourage further exploration of the mechanisms associated with its actions.

## 1. Introduction

Diabetes is a common metabolic disorder characterized by chronic hyperglycemia resulting from defects in insulin production or action [1]. This condition represents a major global health burden, causing approximately 6.7 million deaths annually. Although diabetes has traditionally affected older adults, its prevalence is rising among younger populations [2,3]. The primary causes of type 2 diabetes are insulin resistance, defined by reduced cellular response to insulin, and auto-lipolysis in adipose tissue leading to elevated levels of blood glucose and free fatty acids (FFAs) [4]. Chronic hyperglycemia and elevated FFAs induce apoptosis in pancreatic β-cells via glucolipotoxicity and lipotoxicity, shifting from compensatory hyperinsulinemia to hypoinsulinemia [5,6,7]. Treatment often includes medications such as metformin, which lowers blood glucose by enhancing muscle glucose uptake and reducing hepatic glucose production [8,9]; however, side effects may occur [10]. Natural products rich in antioxidants and anti-inflammatory compounds offer an alternative intervention for diabetes by protecting pancreatic cells from damage, improving their function, and potentially reducing the severity of type 2 diabetes [11,12,13]. Diabetes is increasingly being recognized as a disorder not only of glucose metabolism but also of systemic metabolic and immune homeostasis. Chronic hyperglycemia and insulin resistance are associated with persistent low-grade inflammation and immune dysregulation, which contribute to the development and progression of complications associated with diabetes [14,15]. Recent studies have highlighted that individuals with diabetes often exhibit gut microbial dysbiosis [16,17], which is characterized by a decrease in beneficial bacteria, such as *Akkermansia* and *Lactobacillus*, and by an increase in opportunistic pathogens and pro-inflammatory species [18,19]. These changes can disrupt the integrity of the intestinal barrier, promote systemic inflammation, and exacerbate insulin resistance that worsens metabolic control [20,21].

Given the major role of the gut microbiome in the regulation of the host’s metabolism, immune responses, and energy homeostasis, modulation of the gut microbiome has emerged as a promising therapeutic strategy for the management and prevention of diabetes. Interventions such as dietary modification, prebiotics, probiotics, and plant-derived bioactive compounds have been shown to restore microbial balance, enhance the abundance of short-chain fatty acid (SCFA)-producing bacteria, and reduce inflammation [22,23,24]. These effects can improve insulin sensitivity, promote glucose homeostasis, and protect pancreatic β-cell function [25]. Interventions that target the gut microbiome represent an alternative approach to improving metabolic dysfunction and the immune system in diabetic situations.

*Perilla frutescens*, a functional food plant cultivated for generations in northern Thailand, is traditionally consumed in seed-based dishes. The seeds are nutritionally dense, containing high levels of protein, carbohydrates, and phytochemicals, as do the leaves, with particularly high concentrations of essential fatty acids [26,27,28]. The byproduct of oil extraction, perilla seed residue, is commonly used as animal feed but has been shown to retain significant phytochemicals, including rosmarinic acid, luteolin, and apigenin, comparable to those in the leaves and seeds [29]. These compounds exhibit antioxidant, anti-inflammatory, and metabolic activities, such as enhancing glucose uptake, improving lipid profiles, and supporting insulin signaling [30,31,32]. Based on these properties, perilla seed residue may help prevent type 2 diabetes. However, to date, its potential for diabetes management remains unexplored.

This study aims to investigate the potential of bioactive compounds in the crude extract of perilla seed residue to reduce hyperglycemia by evaluating their effects on β-cell function and the gut microbiome in a type 2 diabetic rat model.

## 2. Results

### 2.1. Perilla Seed Residue Crude Extract (PCE) Characteristics

A total of 8.57 g of PCE was obtained from the ethanol extraction of 100 g of perilla seed residue dried powder. The total phenolic content was 78.02 ± 1.44 mg of gallic equivalent/g extract, and the flavonoid content was 67.63 ± 0.67 mg of catechin equivalent/g extract. HPLC analysis of PCE revealed the presence of rosmarinic acid, luteolin, and apigenin, as shown in Table 1. Antioxidant activity assays showed an antioxidant capacity of 1 g PCE equal to 94.61 ± 2.87 mg ascorbic acid (DPPH), 94.91 ± 5.09 mg Trolox (ABTS), 107.99 ± 9.40 mg Trolox (FRAP), and 668.51 ± 9.85 mg Trolox (ORAC) (Table 2).

### 2.2. Effects of PCE on Growth Parameters in Diabetic Rats

The high-fat diet (HFD) control group demonstrated a progressive increase in body weight, particularly notable by week 10, while food and water intake remained comparable to normal controls. Following STZ injection at week 6, the high-fat diet-induced diabetic rats with streptozotocin (HFD-STZ) exhibited significant body weight loss and increased water intake, while food intake appeared slightly elevated over time compared with controls, indicating diabetic symptoms. Administration of metformin (Met) and both concentrations of PCE (100 and 1000 mg/kg bw) from weeks 7 to 10 resulted in an increase in body weight, reduced water intake, and improved food intake during the final week compared with the HFD-STZ group, as shown in Figure 1. These results underscore the potential of PCE to ameliorate diabetes-induced alterations in growth parameters in this model.

### 2.3. Effects of PCE on Glucose Utilization (FBG Level and OGTT)

Rats in the HFD control group showed a progressive increase in fasting blood glucose (FBG), especially over the last two weeks. In contrast, rats in the HFD-STZ diabetic group exhibited a significant increase in FBG at week 6, measured 72 h after streptozotocin injection (381.10 ± 83.82 mg/dL, *p* < 0.001), compared with the normal control group (118 ± 11.29 mg/dL), confirming the onset of diabetes. Administration of Met and PCE at 100 and 1000 mg/kg bw resulted in a gradual reduction in FBG levels from weeks 7 to 10. Notably, the Met and low-dose PCE groups showed a significant decrease in FBG during the final two weeks. In week 10, the FBG level in the low-dose PCE group was 191.88 ± 45.69 mg/dL, *p* < 0.001 compared with the HFD-STZ group (343.50 ± 35.22 mg/dL), whereas the FBG level of the Met group was 270.88 ± 27.46 mg/dL, *p* < 0.01 (Figure 2A).

The oral glucose tolerance test (OGTT) was used to assess insulin responsiveness and postprandial glucose regulation. While HFD rats showed no significant difference in the AUC compared with normal controls, the HFD-STZ group exhibited a significantly elevated AUC (55,778 ± 3093 mg/dL*min, *p* < 0.001) relative to the normal control group (15,765 ± 367 mg/dL*min). Treatment with low-dose PCE significantly improved glucose tolerance, reducing the AUC to 40,460 ± 4845 mg/dL*min, *p* < 0.001 (Figure 2B,C). Therefore, a low dose of PCE exhibited the potential to improve glucose utilization in HFD-STZ rats.

### 2.4. Effects of PCE on Insulin Secretion and Pathological Appearance of the Pancreas

Although rats in the HFD control group showed no significant differences in fasting blood glucose or OGTT outcomes, there was a trend toward increased insulin secretion and a higher HOMA-IR score (Appendix A), indicating borderline insulin resistance. In contrast, serum insulin levels in the HFD-STZ control group (0.63 ± 0.11 ng/mL, *p* < 0.001) were significantly lower than the normal control group (3.62 ± 1.14 ng/mL), reflecting the effect of STZ on pancreatic destruction. Notably, low-dose PCE treatment significantly increased insulin levels (1.54 ± 0.86 ng/mL, *p* < 0.05), as shown in Figure 3A, whereas no significant difference was observed in the Met and high-dose PCE groups (1.14 ± 0.58 and 1.01 ± 0.42 ng/mL, respectively) when compared with the HFD-STZ control group.

Histological analysis of pancreatic islets revealed that the HFD control group exhibited an increased area and number of islets of Langerhans (Figure 3C,H,I), consistent with enhanced insulin secretion. In contrast, STZ caused destruction of β-cells in the HFD-STZ group compared with the normal control rats, presenting in a small area of islets of Langerhans and β-cell degranulation, as shown in Figure 3B,D. Treatment with PCE at both doses (100 and 1000 mg/kg bw) improved β-cell structure and reduced tissue damage, as shown in Figure 3F,G—results comparable to those following treatment with Met, as shown in Figure 3E. The alteration of pathologic visualization was quantified by measuring the size and counting the number of islets of Langerhans. The HFD-STZ group showed a significant reduction in the size and number of islets of Langerhans compared with the control group (*p* < 0.001). Treatment with PCE at low and high doses could significantly increase islet size and number compared with the HFD-STZ group; this effect was also similar to the results following Met treatment, as shown in Figure 3H,I. These results suggest that PCE effectively preserves pancreatic β-cell structure and function in diabetic rats.

### 2.5. Effects of Perilla Seed Residue on Oxidative Stress and Inflammation

Serum lipid peroxidation, assessed by malondialdehyde (MDA) levels, showed no significant difference between the HFD control group and the normal control group. However, the HFD-STZ control group exhibited a significantly elevated concentration of MDA (2.65 ± 1.01 µM, *p* < 0.001) compared with the normal control group (1.16 ± 0.30 µM), which relates to the reduction of antioxidant capacity in the serum (Figure 4A,B). Treatment with low doses of PCE resulted in the highest level of serum antioxidant capacity and a significant reduction in MDA levels (1.51 ± 0.72 µM) compared with the HFD-STZ group (*p* < 0.01). Therefore, PCE might protect against destruction of the pancreas due to its antioxidant capacity that relates to inflammatory status. To further assess systemic inflammation, serum nitric oxide (NO) levels were measured. As expected, NO levels did not differ significantly between the HFD control and the normal control but were markedly elevated in the HFD-STZ control group. Treatment with low and high doses of PCE resulted in a reduction in serum NO levels in the HFD-STZ rats that were comparable to those of the rats treated with metformin, as shown in Figure 4C. Additionally, inflammatory cytokine levels in the HFD control group were comparable to the normal control group, with the exception of a slight increase in IL-1β (*p* < 0.01), likely due to inflammation-related upregulation of gene expression in adipose tissue, as shown in Figure 4E. In contrast, the HFD-STZ group showed significantly elevated serum levels of TNF-α (304.64  ±  321.60 pg/mL, *p* < 0.05), IL-6 (41.00  ±  26.51 pg/mL, *p* < 0.05), and IL-1β (275.75  ±  85.76 pg/mL, *p* < 0.001) compared with the other controls. Treatment with Met or PCE (both low and high doses) reduced these cytokine levels toward near-control values; however, the reduction was significant only for TNF-α with low-dose PCE (*p* < 0.05), as shown in Figure 4D. It could be suggested that PCE has an anti-inflammatory effect in HFD-STZ rats.

The level of expression of inflammation-related genes in the pancreas was then measured. Rats in the HFD-STZ group exhibited a significantly elevated expression of *TNF-α*, *IL-6*, and *IL-1β*, as well as *iNOS* and *COX-2*, compared with the normal control group. Although treatment with PCE at doses of 100 and 1000 mg/kg bw showed a tendency to reduce the expression of these inflammatory genes, only the reduction in the expression of *iNOS* reached statistical significance relative to the HFD-STZ control group (*p* < 0.001), as shown in Figure 4F.

### 2.6. Effects of PCE on Serum Lipid in HFD-STZ-Induced Diabetic Rats

Serum lipid concentrations were assessed to evaluate glucose and lipid metabolism in HFD and HFD-STZ rats. Overall lipid levels showed no significant differences in HFD rats; however, the HFD-STZ group exhibited significantly elevated triglyceride levels (193.13 ± 66.21 mg/dL) compared with the normal control group (90.13 ± 26.22 mg/dL), which is associated with diabetes. Treatment with both low- and high-dose PCE significantly lowered triglyceride levels, with the greatest reduction observed in the low-dose PCE group (76.62 ± 27.47 mg/dL). Total cholesterol and LDL-C levels were significantly higher in the HFD-STZ group compared with the normal group. Administration of PCE at 100 and 1000 mg/kg bw did not significantly reduce either total cholesterol or LDL-C levels compared with the HFD-STZ group, as shown in Table 3.

### 2.7. Effects of PCE on Diversity of Gut Microbiome

High-quality profiles for fecal samples (*n* = 5 each group) drawn at the study end from normal, HFD, HFD-STZ, HFD-STZ + Met, HFD-STZ + PCE 100 mg/kg bw, and HFD-STZ + PCE 1000 mg/kg bw rats—with a mean read depth of 17,000—were yielded using 16S rRNA V4 sequencing (Illumina; see Section 1). Alpha diversity (Shannon’s diversity index) declined with HFD feeding relative to normal controls (*p* = 0.016) and was further reduced after STZ induction (*p* = 0.056 vs. HFD). Low-dose PCE partially restored diversity (*p* = 0.69 vs. HFD-STZ), whereas high-dose PCE showed an intermediate effect; non-significant increase. PERMANOVA on Weighted UniFrac distances demonstrated clear community separation across groups (pseudo-F = 2.85, R^2^ = 0.112, *p* = 0.007); pair-wise contrasts indicated that both PCE doses shifted microbial composition away from those recorded in HFD-STZ rats and toward normal controls (adjusted *p* < 0.001).

At the genus level, HFD feeding increased *Coriobacteriaceae UCG-002*, *Bilophila*, *Clostridium sensu stricto 1*, and the *Lachnospiraceae NK4A136* group while decreasing the *Eubacterium xylanophilum* group and *Monoglobus* (Figure 5). STZ induction produced additional increases in *DTU014*, *Faecalibaculum*, and *Bacteroides* and further decreases in *Monoglobus, Prevotellaceae UCG-001*, and *Lactobacillus* (Figure 6).

PCE treatment effects: Relative to HFD-STZ controls, both PCE doses resulted in a reduction in DTU014, and other taxa were enriched in diabetic rats, with low-dose PCE resulting in the largest effect. PCE also increased beneficial genera implicated in gut barrier and metabolic regulation, including *Lactobacillus*, *Clostridium sensu stricto 1* (beneficial clades), and *Akkermansia*; responses were generally greater in rats given a low dose of PCE than those given a high dose of PCE (Figure 7, Figure 8 and Figure 9). Direction and magnitude for *Bacteroides* (including *B. fragilis*) were inconsistent across analyses; reanalysis was indicated.

Full differential abundance statistics for all detected taxa are provided in Appendix A.

At the genus taxonomic level, the HFD and HFD + STZ groups exhibited reduced diversity in the gut microbiome, dominated by fewer bacterial genera such as *Faecalibaculum*. In contrast, the PCE-treated groups (100 and 1000 mg/kg bw) showed a more balanced microbial composition, closely resembling that of the non-treated control group. Beneficial genera such as *Blautia*, along with partial restoration of microbial diversity, were notably increased in both the treatment and the control groups, suggesting that PCE contributes to the reestablishment of gut bacterial balance disrupted by STZ (Figure 8).

Analysis of 155 bacterial genera, focusing on the top six showing significant population differences, revealed that the HFD-STZ control group also exhibited an increased abundance of the *DTU014*, *Negativibacillus*, and *Eubacterium nodatum* groups. However, the *Negativibacillus* and *Eubacterium nodatum* groups showed no statistical difference compared with normal controls. Both PCE doses (100 and 1000 mg/kg bw) reduced these populations in a dose-dependent manner relative to HFD-STZ controls (Figure 9A,C). Concurrently, PCE treatment enhanced beneficial genera, including *Lactobacillus*, *Clostridium sensu* stricto 1, and *Akkermansia*, with the low dose of PCE showing greater efficacy, as shown in Figure 9E,G. Further species-level metagenomic analysis (*n* = 294 species) revealed two species with notable intergroup differences. Specifically, there was a significant reduction in pathogenic species, including *Bacteroides fragilis*, and in butyrate-producing bacteria in the HFD-STZ control rats. Both PCE dosages ameliorated these dysbiotic changes by decreasing the abundance of pathogenic taxa while increasing beneficial butyrate producers (Figure 9H,J). Collectively, these findings indicate that PCE supplementation results in the restoration of a more balanced and health-associated composition of the gut microbiome in this model.

A heatmap correlation analysis was performed to clarify the link between gut microbiome changes and metabolic outcomes. HFD and HFD-STZ rats disrupted gut microbiome correlations, strengthening links between harmful bacteria and adverse metabolic markers while weakening associations with beneficial genera. PCE treatment at both doses partially reversed these effects, restoring negative correlations between beneficial genera (e.g., *Akkermansia*, *Muribaculaceae*) and metabolic markers, highlighting the role of PCE in metabolic improvement via gut microbiome modulation (Appendix A).

## 3. Discussion

Bioactive substances such as phenolic acids and flavonoids are abundant in the fruits and seeds of *Perilla frutescens*. Even after oil extraction, the seed residue retains over 72% of key phytochemicals such as luteolin, apigenin, and rosmarinic acid that have anti-inflammatory and antioxidant properties [29,30,33]. The amounts of these compounds that are extracted can be increased by using higher ethanol concentrations, [34]. This demonstrates the potential of perilla for useful products and health-related uses. Rosmarinic acid enhances the uptake of GLUT4-mediated glucose, inhibits hepatic gluconeogenesis via PEPCK, and reduces the production of pro-inflammatory cytokines and mediators [35,36,37]. In addition to these metabolic effects, it modulates the gut microbiome by increasing beneficial taxa (e.g., *Lactobacillus johnsonii*, *Candidatus Arthromitus*, SCFA producers) and by reducing potential pathogens such as *Escherichia coli* and *Bifidobacterium pseudolongum* in rats [38]. Apigenin improves lipid profiles and suppresses inflammatory markers [31,39] and beneficially alters the gut microbiome in disease models by increasing beneficial bacteria (e.g., *Akkermansia*), reducing harmful taxa, and strengthening the intestinal barrier—which protects against inflammation and pathogens [40]. Luteolin also lowers serum triglycerides and LDL-C, suppresses inflammation, upregulates Irs2 to improve insulin signaling [41,42,43], and promotes a healthier gut microbiome and barrier function, reducing systemic inflammation [44]. The luteolin, apigenin, and rosmarinic acid found in perilla seed residue enhance insulin signaling, lower inflammation, and improve glucose metabolism and the gut microbiome [35,37,39]. These effects indicate that the residue may be effective in treating type 2 diabetes, warranting its testing in appropriate animal models.

For the study of type 2 diabetes, animal models are crucial [43]. Rats fed a high-fat diet (HFD) show early insulin resistance, which is typified by weight gain, increased fat accumulation, and compromised insulin and glucose regulation. Long-term insulin resistance causes glucotoxicity and lipotoxicity, which induce progressive β-cell dysfunction [45,46]. Streptozotocin (STZ), which selectively destroys pancreatic β-cells, is used to simulate β-cell damage and insulin insufficiency [47,48]. This results in severe hyperglycemia and decreased insulin secretion. In this study, STZ injections in HFD-fed rats (HFD-STZ group) effectively produced diabetic symptoms, which closely resemble the pathophysiology of type 2 diabetes, according to Sakuludomkan W. et al. (2020) [49]. These symptoms, including polydipsia, weight loss, reduced insulin levels, impaired glucose tolerance, and pancreatic islet damage, were also observed in our experimental results. The destruction of β-cells and decreased insulin production are closely associated with pancreatic inflammation [48]. Supporting this, diabetic rats exhibited elevated levels of systemic inflammatory markers and an increased expression of pro-inflammatory genes (*TNF-α*, *IL-1β*, *IL-6*, *iNOS,* and *COX-2*), which contribute to a decrease in islet size and number in the pancreas. Additionally, chronic hyperglycemia, along with an increase in triglycerides and total cholesterol, induces dyslipidemia, disrupts lipid metabolism, and elevates cardiovascular risk [50]. Therapeutic strategies that lower blood glucose and restore β-cell function by mitigating pancreatic inflammation may help alleviate diabetic symptoms. Therefore, in this study, the potential benefits of perilla seed residue crude extract were investigated by assessing its effects in diabetic rats.

*Perilla frutescens* seed residue crude extract (PCE) dramatically reduced the symptoms of diabetes in diabetic rats that were induced by a high-fat diet and streptozotocin (HFD-STZ). At low doses, PCE treatment resulted in a reduction of over 43% in blood glucose levels, decreased excessive water intake, and encouraged weight gain. It most likely lowers blood sugar by suppressing hepatic gluconeogenesis, improving pancreatic β-cell function by reducing inflammation, and may increase glucose uptake through GLUT4 via the phytochemicals in PCE [35,36,39]. Similar to metformin treatment, PCE also restored the size and number of pancreatic islets and more than tripled insulin secretion. Furthermore, by lowering levels of oxidative stress and systemic nitric oxide and downregulating pro-inflammatory cytokines, PCE demonstrated potent anti-inflammatory effects. It also enhanced lipid profiles by significantly reducing triglycerides. It is noteworthy that the lower dose exerted stronger anti-diabetic effects than the higher doses in experimental rats. This finding may suggest that certain components, when present at higher concentrations, exert inhibitory effects. Because we used a crude extract, the precise composition of both agonistic and antagonistic compounds remains unclear. The most pronounced anti-diabetic effect was observed at 100 mg/kg bodyweight, indicating that this dose may provide an optimal balance of phytochemicals after metabolism in rats and may explain why higher doses did not enhance efficacy.

The gut microbiome plays a crucial role in metabolic health, with type 2 diabetes often associated with dysbiosis marked by a decline in beneficial microbes and an increase in opportunistic taxa. Environmental factors, aging, and diet can disrupt microbial balance, thereby affecting insulin sensitivity, inflammation, and metabolism [51,52,53]. A high-fat diet (HFD) disrupts gut microbial homeostasis in rats, promoting pathogenic bacterial taxa while depleting beneficial populations. This dysbiosis impairs glucose regulation, and, over time, HFD consumption may eventually lead to the development of diabetes [54,55]. For instance, STZ-induced diabetes produced a gut microbiome profile similar to HFD, but with notable differences such as a significant increase in *Bacteroides* abundance. This increase was positively correlated with heightened predicted lipopolysaccharide (LPS) biosynthesis, a process known to promote systemic inflammation and insulin resistance, both of which are hallmark characteristics of diabetes pathophysiology [56]. Conversely, *Lactobacillus* abundance declined, potentially due to elevated *Faecalibaculum* levels, which are known to inhibit *Lactobacillus* growth [57]. *Lactobacillus* exhibits notable anti-diabetic and antioxidant effects in rats, highlighting its probiotic potential for improving glucose metabolism and mitigating diabetic complications [58].

These compositional and functional changes in the gut microbiome may influence the production of short-chain fatty acids (SCFAs), such as butyrate, which play a role in glucose metabolism and insulin sensitivity. Diabetic individuals often have lower levels of SCFA-producing bacteria. Probiotic species such as *Lactobacillus*, which supports glycemic control, and *Akkermansia muciniphila*, which enhances insulin sensitivity and reduces inflammation, are of particular interest in diabetes research. Consistent with previous reports [23,53,59], rats in the HFD-STZ diabetic group showed reduced microbial diversity and a distinct microbial community profile compared with healthy controls. Alpha diversity analysis revealed a non-significant downward trend at the generic level, potentially due to sample size or sequencing limitations. Beta diversity analysis clearly separated the microbial communities of both the diabetic and the control rats, indicating substantial differences in community structure. Diabetes was associated with an increased abundance of genera such as *Blautia*, *Muribaculaceae,* and *Lachnospiraceae.* These findings underscore that, even when diversity metrics are not statistically significant, compositional shifts in the microbiome are biologically meaningful. Therefore, comprehensive community analysis is essential for understanding microbiome alterations linked to inflammation and insulin resistance [60].

PCE treatment notably shifted the gut microbiome toward a healthier state, increasing microbial richness and diversity and aligning the community profile with non-diabetic controls, as reflected in beta diversity analysis. Taxonomic changes included an increase in beneficial bacteria, consistent with its rich content of polyphenols, which can serve as substrates for gut bacteria and selectively promote the growth of health-associated taxa, similar to established prebiotics like inulin [53]. The lower dose (100 mg/kg bw) had greater effects on both the microbiome and metabolic outcomes, suggesting optimal dosing for prebiotic benefits. Overall, these results support the prebiotic potential of PCE and highlight the importance of further research using larger cohorts and metagenomic analysis to detail its microbiome interactions. PCE treatment notably enriched gut bacterial taxa involved in SCFA production and metabolic regulation. Significant increases were observed in *Bacteroides fragilis*, known for fermenting complex carbohydrates into acetate and propionate and producing immunomodulatory polysaccharide A. These changes are associated with improved insulin sensitivity, reduced gut inflammation, and enhanced glycemic control [61,62,63]. Similarly, the genus *DTU014* (phylum Firmicutes) was elevated, a group linked to metabolites that influence glucose metabolism [64,65]. Treatment with PCE also increased *Lactobacillus* species, which are widely recognized for promoting gut health and reducing insulin resistance [53,66]. Interestingly, low-dose PCE led to even higher *Lactobacillus* levels, mirroring its superior glucose-lowering benefits. Moreover, *Clostridium sensu stricto 1* and its butyrate-producing members, were also enriched, supporting improved gut barrier integrity, increased GLP-1 secretion, and better glycemic control [53,67,68]. PCE treatment also significantly increased *Akkermansia muciniphila*, a key gut species linked to barrier integrity and reduced inflammation. This boost was accompanied by lower inflammatory markers, enhanced insulin secretion, and better lipid profiles [63,69,70]. Collectively, these taxonomic shifts demonstrate that PCE enhances SCFA production, strengthens the gut barrier, and suppresses inflammation, providing its positive effects on glucose homeostasis and metabolic health in diabetic rats. Alongside increasing beneficial microbes, PCE treatment suppressed several taxa linked to inflammation and metabolic dysfunction in diabetes. Specifically, *Negativibacillus* and the *Eubacterium nodatum* group, both elevated in HFD-STZ rats and associated with inflammation and poor glycemic control [53,65,70], were significantly reduced by PCE, approaching levels seen in healthy controls. This reduction likely results from the impact of PCE on enhancing the production of SCFA and the lowering of gut pH, creating an environment unfavorable for opportunistic, pro-inflammatory bacteria. Although differences between these taxa and normal controls were not always statistically significant, their consistent decline after treatment with PCE aligns with observed decreases in inflammatory markers (TNF-α, IL-1β, IL-6) and improved insulin signaling. Overall, by selectively suppressing pathogenic and inflammatory microbes while promoting SCFA-producing bacteria, PCE contributes to reduced inflammation and improved metabolic health [71,72], supporting a healthier gut environment in diabetic rats. PCE treatment induced changes in the microbiome similar to those of metformin and other prebiotic therapies, notably increasing beneficial SCFA-producing genera such as *Akkermansia* and reducing inflammatory taxa [63]. These shifts were associated with improved glycemic control, enhanced gut barrier function, and reduced inflammation in diabetic rats, outcomes that were on a par with or superior to metformin. As a natural, polyphenol-rich supplement, PCE may provide metabolic and anti-inflammatory benefits with fewer side effects. These results suggest PCE is a promising prebiotic adjunct to diabetes therapy, improving hyperglycemia in diabetic rats by increasing gut microbial diversity, boosting SCFA-producing bacteria, and reducing inflammation. Further studies are needed to confirm these benefits in humans and determine optimal use.

PCE treatment shows promise in alleviating diabetes by enriching beneficial gut bacteria and suppressing harmful taxa. Interestingly, although the low dose resulted in greater microbial diversity, it also led to a more pronounced increase in bacteria linked to glucose regulation. These microbial shifts correspond with observed reductions in blood glucose and improvement of diabetic symptoms, as summarized in Figure 10. Further studies are warranted to identify the optimal dosage for maximal therapeutic impact.

## 4. Materials and Methods

### 4.1. Plant Material and Extraction

*Perilla frutescens* (Nga-kee-mon) seeds were collected from Baan San Khong, Dok Kham Tai District, Phayao, Thailand, and the specimens were deposited at the Queen Sirikit Botanic Garden Herbarium, Chiang Mai (Code: QBG-93756). After oil was extracted from the seeds by mechanical pressing, the resulting dried seed residue was blended and extracted with 80% ethanol (1:10 *w*/*v*) overnight at room temperature. The extract was then filtered, evaporated, and lyophilized to yield perilla seed residue crude extract (PCE) powder.

### 4.2. Measurement of Total Phenolic Content

The total phenolic content in PCE was measured by the Folin–Ciocalteu assay as described by Tantipaiboonwong et al. (2017) [73]. Briefly, 20 µL of PCE, 80 µL of 7.5% sodium carbonate, and 100 µL of 10% Folin–Ciocalteu reagent were mixed together. After incubating for 30 min at room temperature, absorbance was read at 765 nm. Total phenolics were quantified using a gallic acid standard curve and expressed as mg gallic acid equivalent (GAE) per gram of extract. Three independent experiments were performed in all cases, and mean values were calculated.

### 4.3. Measurement of Total Flavonoid Content

The total flavonoid content in PCE was determined using the aluminum chloride colorimetric assay as described by Tantipaiboonwong et al. (2017) [73]. Briefly, 25 µL of PCE was mixed with 7.5 µL of 5% sodium nitrite and 125 µL of distilled water and then incubated for 6 min at room temperature. After this, 15 µL of 10% aluminum chloride was added and incubated for 5 min, followed by the addition of 50 µL of 1 M sodium hydroxide and 27.5 µL of distilled water. Absorbance was measured at 510 nm, and flavonoid content was quantified using a catechin standard curve, expressed as milligrams of catechin equivalent (CE) per gram of extract. Three independent experiments were performed in all cases, and mean values were calculated.

### 4.4. Determination of ABTS and DPPH by Free Radical Scavenging Assay

The ABTS scavenging ability of PCE was determined in accordance with the method of Brand-Williams, Cuvelier, and Berset (1995) [74] and expressed as Trolox equivalents per gram of extract. The DPPH radical scavenging activity was also evaluated using the same method, with results expressed as ascorbic acid equivalent per gram of extract. Three independent experiments were performed in all cases, and mean values were calculated.

### 4.5. Ferric Reducing Antioxidant Power (FRAP) Assay

The FRAP of PCE was determined using the colorimetric assay described by Benzie and Strain (1996) [75]. This method measures the reduction of ferric ions to the blue ferrous form in the Fe(TPTZ)_2_Cl_3_ complex under acidic conditions. Briefly, 20 µL of each sample was mixed with 180 µL of FRAP reagent and incubated for 5 min at 37 °C. Absorbance was read at 595 nm. Total antioxidant capacity was calculated from a Trolox standard curve and reported as mg Trolox equivalent (TE) per gram of extract. Three independent experiments were performed in all cases, and mean values were calculated.

### 4.6. Oxygen Radical Absorbance Capacity (ORAC) Assay

Following the established protocol of Davalos et al. (2004) [76], the assay was conducted in a 75 mM phosphate buffer (pH 7.4) to a total reaction volume of 200 µL. Antioxidant solution (25 µL) and 25 nM Fluorescein (150 µL) were added to each well of a microplate and preincubated for 10 min at 37 °C. AAPH solution (25 µL; 1.56 M final concentration) was then rapidly added using a multichannel pipette. Fluorescence was recorded every minute for 180 min with automatic shaking before each reading. A blank control containing a phosphate buffer instead of antioxidant as well as calibration solutions with Trolox (0.625–10 µM) were included in each assay. All reactions were performed in duplicate, with at least three independent assays per sample. ORAC-PE values were calculated as described for ORAC-FL assays. Three independent experiments were performed in all cases, and mean values were calculated.

### 4.7. Measurement of Known Active Compounds in PCE by HPLC

The phenolic compounds in PCE were identified by HPLC following the method of Pintha et al. (2014) [77]. PCE was dissolved in methanol at 5 mg/mL and filtered through a 0.2 µm membrane. Separation was performed with an ODS-3-C18 column (4.6 × 250 mm, 5 µm; Agilent, Santa Clara, CA, USA) using a mobile phase of 0.1% trifluoroacetic acid in water (phase A) and methanol (phase B). The gradient elutions were: 90–10% A and 10–90% B from 0 to 35 min and then 10–90% A and 90–10% B from 35 to 40 min, at a flow rate of 1.0 mL/min. Phenolic compounds were detected at 320 nm. Peak areas were quantified against standards of rosmarinic acid (Abcam, Cambridge, UK), luteolin, and apigenin (Chengdu Biopurify Phytochemical Ltd., Chengdu, China). Two independent experiments were performed with two duplicates.

### 4.8. High-Fat Diet Production and Characterization

High-fat diets are often more energy dense and may be less effective at inducing satiety, potentially leading to increased caloric consumption. In this study, the normal diet (4.5% energy from fat) and high-fat (50–55% energy from fat) diets were prepared in-house by combining a commercial basal diet with lard, vegetable oil, and margarine.

### 4.9. Animal and Experimental Protocol

Forty-eight eight-week-old male Wistar rats were sourced from Nomura Siam International Co., Ltd., Pathumwan, Bangkok, Thailand and acclimatized at the Faculty of Medicine, Chiang Mai University, in an air-conditioned environment maintained at 25 °C with a 12:12 h light/dark cycle. Animals had free access to a standard diet and water. All procedures were approved by the Faculty of Medicine, Chiang Mai University Animal Ethics Committee (approval no. 02/2567).

Eight rats in group 1 were fed a normal diet, while group 2 was fed an HFD throughout the experiment to induce diabetes. Thirty-two rats (groups 3–6) were randomly allocated into 4 groups, and insulin resistance was induced with a high-fat diet for 6 weeks. Groups 3–6 were then injected with streptozotocin (STZ) dissolved in a citrate buffer at pH 4.5 via at a concentration of 35 mg/kg bw in order to impair the ability of the pancreatic beta cells to secrete insulin, resulting in elevated blood sugar levels. Blood sugar levels were then measured after 72 h. Rats with blood sugar levels higher than 200 mg/dL were considered to demonstrate diabetes. During weeks 7 to 10, the rats in groups 1 and 2 received 10% DMSO as a control solvent of PCE. Group 4 received metformin (Met) at 100 mg/kg bw as a positive control. Groups 5 and 6 were orally administered PCE daily at 100 mg/kg bw and 1000 mg/kg bw for four weeks, with all groups except group 1 being continued on a high-fat diet until the end of the experiments. Body weight, food, and water intake were measured weekly. Fasting blood glucose was monitored with a glucometer (OK Biotech, model Easy G, Taiwan; strip pad lot number: S231228D-1) at baseline (week 0), weeks 1, 3, and 5, and weekly throughout PCE administration (weeks 6–10), as shown in Figure 11.

### 4.10. Euthanasia and Tissue Collection

At the end of the treatment, rats were fasted for 12 h and euthanized under deep anesthesia with di-isoflurane. Blood was collected from the hepatic vein and centrifuged at 3500× *g* for 10 min. Serum was used to analyze antioxidant levels, oxidative stress, lipid profile, and inflammation. Adipose tissue was collected for inflammation assessment. Pancreatic tissue was divided, with one portion stored at –80 °C and the other fixed in formalin for analysis of β-cell pathology and inflammation. Fecal samples were collected prior to euthanasia in sterile tubes containing DNA/RNA preservation solution for sequencing.

### 4.11. Glucose Tolerance Measurement

An oral glucose tolerance test (OGTT) was performed to assess glucose utilization and insulin function. Rats were fasted overnight for 12 h, and baseline blood glucose was assessed at time 0 from 10 µL of tail vein blood using a glucometer (OK Biotech, model Easy G, Taiwan; strip pad lot number: S231228D-1, 95.5% accuracy). Rats were then given an oral glucose dose (2 g/kg body weight, 2 g/mL solution), and blood glucose levels were recorded at 30, 60, 90, and 120 min post administration. A glucose concentration–time curve was plotted, and the area under the curve (AUC) was calculated for each rat.

### 4.12. Fasting Serum Insulin Level

Serum was analyzed for insulin concentration using an enzyme-linked immunosorbent assay (ELISA) kit in accordance with the manufacturer’s instructions (Merck Millipore, Darmstadt, Germany). The fasting insulin level was calculated from a standard curve of insulin (ng/mL).

### 4.13. Examination of Pancreatic Beta Cell Pathology

Formalin-fixed pancreatic tissue was embedded in paraffin and sectioned for slide preparation. Hematoxylin and eosin (H&E)-stained slides, prepared by the Department of Pathology, Faculty of Medicine, Chiang Mai University, were used to assess β-cell pathology. The number and total area of islets of Langerhans were quantified and analyzed alongside insulin production. The aim of these results was to investigate the restoration of pancreatic structure and function following PCE treatment compared with the HFD-STZ control group.

### 4.14. Biochemical Analysis (Lipid Profile, MDA, Nitric Oxide)

Total triglyceride and cholesterol content was measured using automated analyzers (Sysmex BX-3010, Shiojiri, Japan) at the Small Animal Hospital, Chiang Mai University. Serum malondialdehyde (MDA) levels were assessed by the TBARS assay as described by Chaiyasut et al. (2011) [78]. Serum nitric oxide (NO) production was measured using the Griess reaction following the method of El-Shakour et al. (2015) [79].

### 4.15. Measurement of Inflammatory Cytokines in Serum

The concentrations of IL-6, IL-1β, and TNF-α in rat serum were measured using an ELISA kit (Elabscience, Houston, TX, USA) in accordance with the manufacturer’s instructions. Serum samples (100 μL) were added to antibody-coated microplates and incubated at 37 °C. After sequential incubations with biotinylated detection antibody and HRP-conjugated streptavidin, color development was achieved with a TMB substrate. Absorbance was measured at 450 nm and normalized with negative controls.

### 4.16. Determination of Level of Expression of Inflammatory Cytokines

Approximately 50 mg of pancreatic tissue or 500 mg of adipose tissue were homogenized in 1 mL of TRIzol reagent and incubated at room temperature for 10 min. After the addition of chloroform and after centrifugation, the aqueous phase was collected, and RNA was precipitated with isopropanol, centrifuged at 12,000× *g* for 10 min at 4 °C, washed, dried, and dissolved in DEPC-treated water. RNA purity and concentration were measured using a NanoDrop spectrophotometer (Thermo Fisher Scientific, Waltham, MA, USA). Complementary DNA (cDNA) was synthesized using ReverTra Ace qPCR RT Master Mix (TOYOBO, Osaka, Japan) in line with the manufacturer’s instructions. Real-time PCR was performed in a 20 µL reaction containing 1 µg cDNA, 2× SYBR Green/ROX qPCR Master Mix, gene-specific primers (*IL-6*, *IL-1β*, *TNF-α*, *iNOS*, *COX-2*, and *GAPDH* as reference), and DEPC-treated water using a QuantStudio™ 6 Flex system (Thermo Scientific, Waltham, MA, USA). The cycling conditions were 95 °C for 10 min, followed by 40 cycles of 95 °C for 15 s and 60 °C for 60 s. Primer sequences were as follows: *TNF-α* (F: 5′-AAATGGGCTCCCTCTCATCAGTCC-3′, R: 5′-TCTGCTTGGTGGTTTGCTACGAC-3′), *IL-6* (F: 5′-TCCTACCCCCAACTTCAATGCTC-3′, R: 5′-TTGGATGGTCTTGGTCCTTAGCC-3′), *IL-1β* (F: 5′-CACCTCTCAAGCAGAGCACAG-3′, R: 5′-GGGTTCCATGGTGAAGTCAAC-3′), *iNOS* (F: 5′-CATTGGAAGTGAAGCGTTTCG-3′, R: 5′-CAGCTGGGCTGTACAAACCTT-3′), *COX-2* (F: 5′-GCCCACCAACTTACAATGTGC-3′, R: 5′-CATGGGAGTTGGGCAGTCAT-3′), and *GAPDH* (F: 5′-GACATGCCGCCTGGAGAAAC-3′, R: 5′-AGCCCAGGATGCCCTTTAGT-3′). Gene expression was normalized to *GAPDH* and expressed as relative C_t_ values compared with controls.

### 4.17. Gut Microbiome Analysis

The composition of the gut microbiome in the fecal samples was assessed via 16S rRNA amplicon sequencing at the Pilot Plant Development and Training Institute, King Mongkut’s University of Technology Thonburi. The V4 hypervariable region was amplified with primers 515F (5′-GTGCCAGCMGCCGCGGTAA-3′) and 806R (5′-GGACTACHVGGGTWTCTAAT-3′). Paired-end sequencing (2 × 250 bp) was performed on an Illumina NovaSeq 6000, San Diego, CA, USA following the manufacturer’s protocol. Raw sequences were quality filtered, merged, and chimera-checked. High-quality sequences were taxonomically classified against the SILVA database v138 using QIIME2 (2021.8.0). Alpha diversity (Shannon’s diversity index) and beta diversity (Weighted Unifrac distances) were calculated and visualized in R (v2024.09.1+394).

### 4.18. Statistical Analysis

In vivo data are presented as mean ± standard deviation (SD) and were calculated using Microsoft Excel. Statistical analysis of differences between groups was performed using one-way ANOVA (GraphPad Prism software, version 9.5.1, San Diego, CA, USA). Statistical significance was established at thresholds of *p* < 0.05 or *p* < 0.01. Variations in alpha diversity among groups were assessed using the Kruskal–Wallis test. Beta diversity was evaluated by PERMANOVA employing Weighted UniFrac distances, with statistical significance defined as *p* < 0.05.

## 5. Conclusions

In conclusion, phytochemicals from perilla seed residue lower fasting blood glucose in diabetic rats by enhancing insulin sensitivity via anti-inflammatory effects, protecting β-cells from the oxidative stress caused by a high-fat diet and streptozotocin, and reducing diabetes-related complications. Notably, these effects were dose independent, with optimal benefits observed at 100 mg/kg bw. PCE also results in the restoration of beneficial gut bacteria (e.g., *Lactobacillus*, *Clostridium sensu stricto 1*, *Akkermansia*) while reducing pathogenic taxa (e.g., *Negativibacillus*, *Eubacterium nodatum group*), indicating improved metabolic health through microbiome rebalancing and reduced inflammation. These findings highlight the potential of perilla-derived products as natural anti-diabetic agents and underscore their value in the field of Thai medicinal herbs.

## Figures and Tables

**Figure 1 ijms-26-08176-f001:**
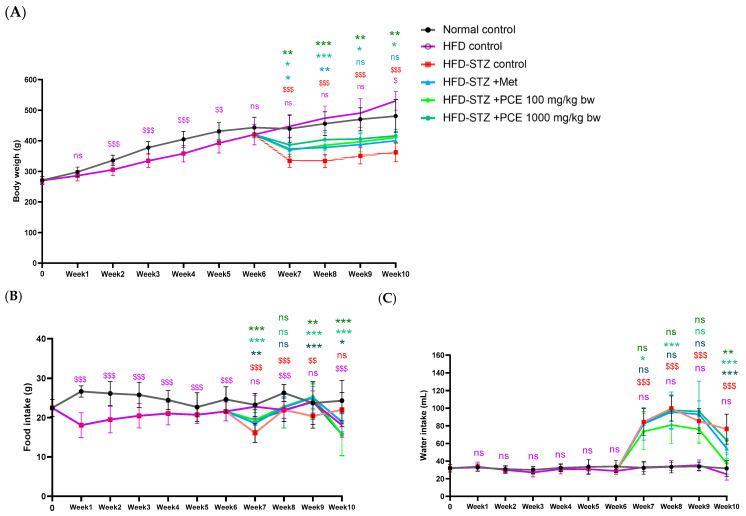
Effects of PCE on diabetic rat characteristics presented as time against body weight (**A**), food intake (**B**), and water intake (**C**) in diabetic rats. Results are presented as mean ± SD (*n* = 8). ns, not significant. * *p* < 0.05, ** *p* < 0.01, and *** *p* < 0.001 compared with HFD-STZ control. ^$^ *p* < 0.05, ^$$^ *p* < 0.01, and ^$$$^ *p* < 0.001 compared with normal control.

**Figure 2 ijms-26-08176-f002:**
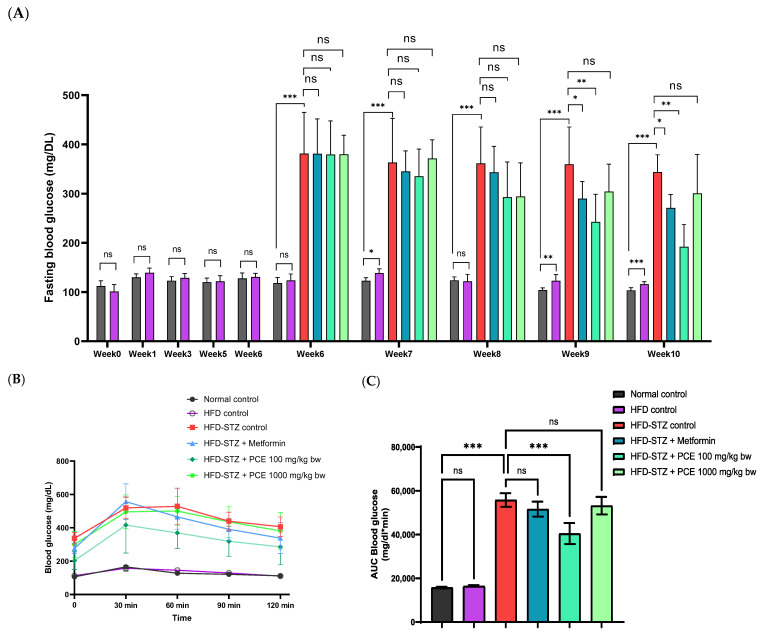
Effects of PCE on glucose utilization. Presented as fasting blood glucose in diabetic rats (**A**), glucose tolerance—as shown by oral glucose tolerance test curves (**B**), and area under the curve (**C**). Results are presented as mean ± SD (*n* = 8). ns, not significant. * *p* < 0.05, ** *p* < 0.01, and *** *p* < 0.001 compared with HFD-STZ control.

**Figure 3 ijms-26-08176-f003:**
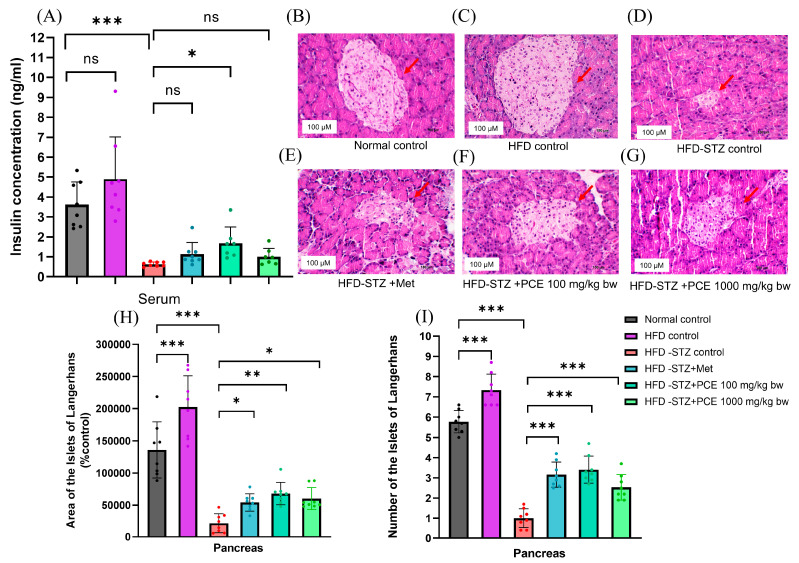
Effects of PCE on pancreatic toxicity induced by streptozotocin. Insulin levels (**A**) and histopathology of islets of Langerhans in the following groups: normal control (**B**), HFD control (**C**), HFD-STZ (**D**), HFD-STZ + Metformin (**E**), HFD-STZ + PCE 100 mg/kg bw (**F**), and HFD-STZ + PCE 1000 mg/kg bw (**G**). Quantitative analysis of the number of islets of Langerhans (**H**) and their calculated area (**I**). Results are presented as mean ± SD (*n* = 8). ns, not significant. * *p* < 0.05, ** *p* < 0.01, and *** *p* < 0.001 compared with HFD-STZ control.

**Figure 4 ijms-26-08176-f004:**
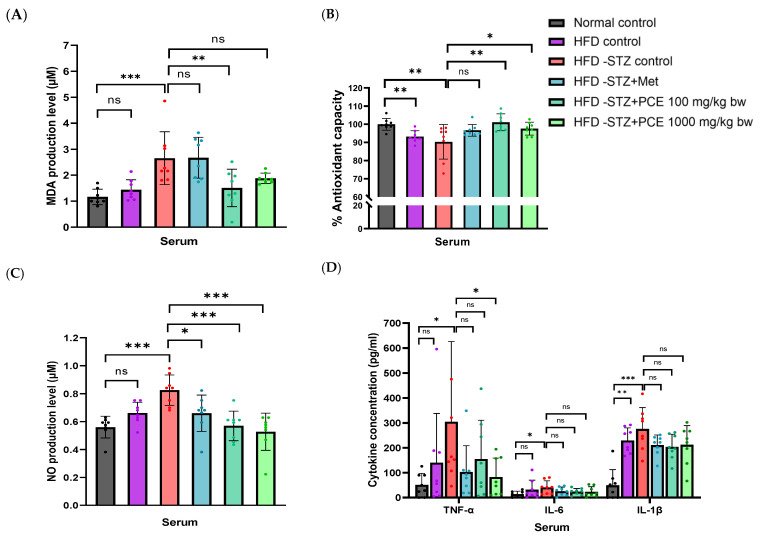
Effects of PCE on inflammatory and oxidative stress markers. Systemic oxidative stress measured by malondialdehyde (MDA) production (**A**) and percentage of antioxidant capacity in serum (**B**). Systemic inflammation indicated by serum nitric oxide (NO) (**C**) and cytokine levels (**D**). Fold changes of *TNF-α*, *IL-6*, *IL-1β*, *iNOS*, and *COX-2* in adipose tissue (**E**) and pancreatic expression (**F**). Results are presented as mean ± SD (*n* = 8). ns, not significant. * *p* < 0.05, ** *p* < 0.01, and *** *p* < 0.001 compared with HFD-STZ control.

**Figure 5 ijms-26-08176-f005:**
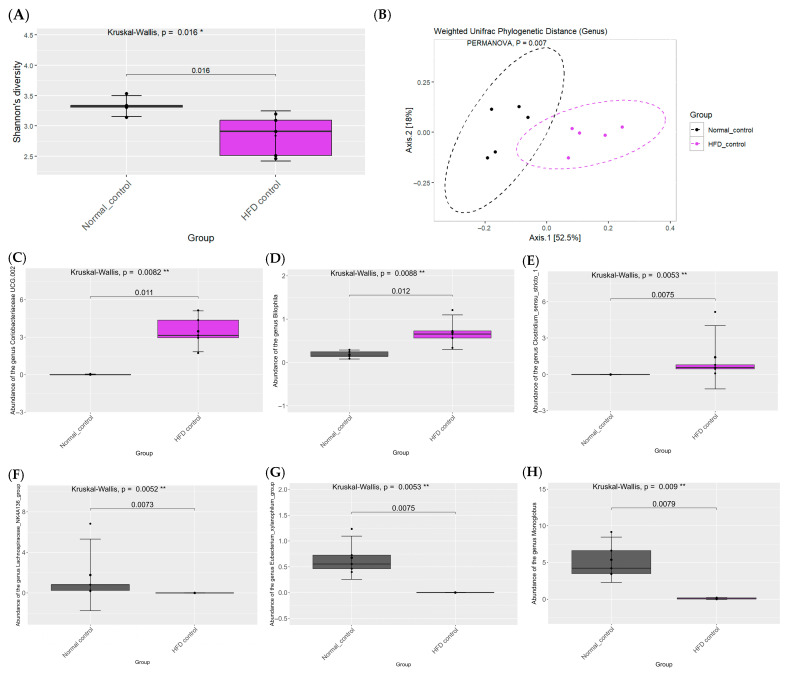
Effects of high-fat diet on gut microbiome in rats. Alpha diversity at the genus level (**A**). Beta diversity at the genus level (**B**). *Coriobacteriaceae UCG-002* (**C**). *Bilophila* (**D**). *Clostridium sensu stricto 1* (**E**). *Lachnospiraceae NK4A136* group (**F**). *Eubacterium xylanophilum* group (**G**). *Monoglobus* (**H**). Group differences were evaluated using the Kruskal–Wallis test to determine statistical significance. ** p* < 0.05, and *** p* < 0.01 was used to indicate a significant difference. The number of rats was 5 per group.

**Figure 6 ijms-26-08176-f006:**
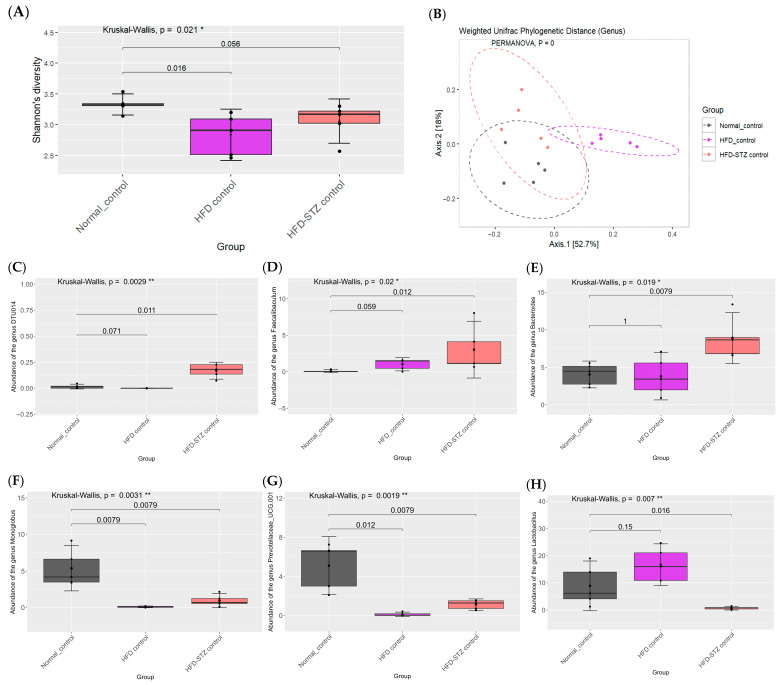
Effects of high-fat diet and STZ on gut microbiome in diabetic rats. Alpha diversity at the genus level (**A**). Beta diversity at the genus level (**B**). *DTU014* (**C**). *Faecalibaculum* (**D**). *Bacteroides* (**E**). *Monoglobus* (**F**). *Prevotellaceae UCG-001* (**G**). *Lactobacillus* (**H**). Group differences were evaluated using the Kruskal–Wallis test to determine statistical significance. * *p* < 0.05, and *** p* < 0.05 was used to indicate a significant difference. The number of rats was 5 per group.

**Figure 7 ijms-26-08176-f007:**
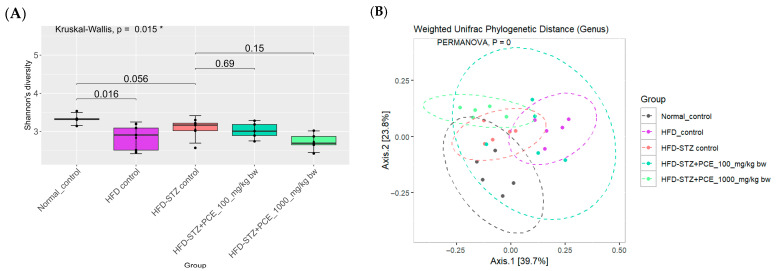
Effects of PCE on gut microbiome in diabetic rats. Alpha diversity at the genus levels (**A**). Beta diversity at the genus level (**B**). Group differences were evaluated using the Kruskal–Wallis test to determine statistical significance. ** p* < 0.05 was used to indicate a significant difference. The number of rats was 5 per group.

**Figure 8 ijms-26-08176-f008:**
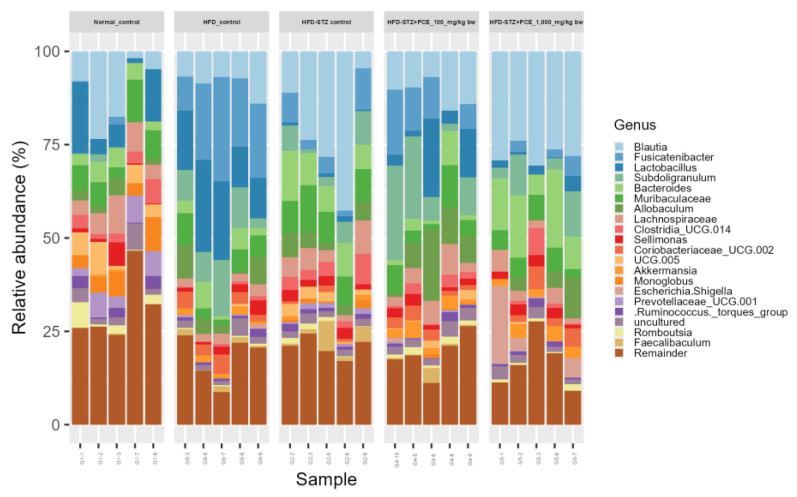
Effects of PCE on percent relative abundances of the taxonomic compositions in diabetic rats at the genus level. The number of rats was 5 per group.

**Figure 9 ijms-26-08176-f009:**
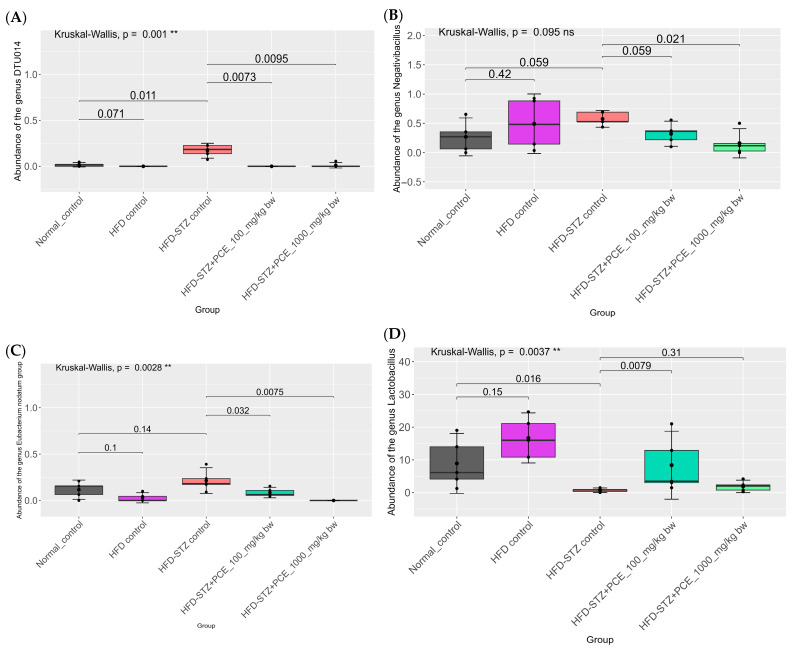
Effects of PCE on relative abundance of bacterial genera and selected species in diabetic rats. Relative abundances of *DTU014* (**A**), *Negativibacillus* (**B**), the *Eubacterium nodatum* group (**C**), *Lactobacillus* (**D**), *Clostridium sensu stricto 1* (**E**), *Akkermansia* (**F**), butyrate-producing bacteria (**G**), and *Bacteroides fragilis* (**H**). Group differences were evaluated using the Kruskal–Wallis test to determine statistical significance. ns, not significant. ** *p* < 0.01, and *** *p* < 0.001 was used to indicate a significant difference. The number of rats was 5 per group.

**Figure 10 ijms-26-08176-f010:**
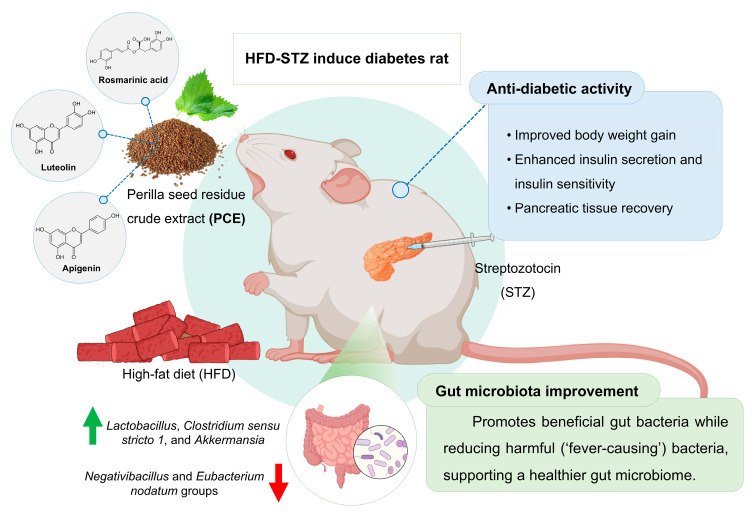
Proposed summary of action of Perilla seed residue on induced diabetic rat model.

**Figure 11 ijms-26-08176-f011:**
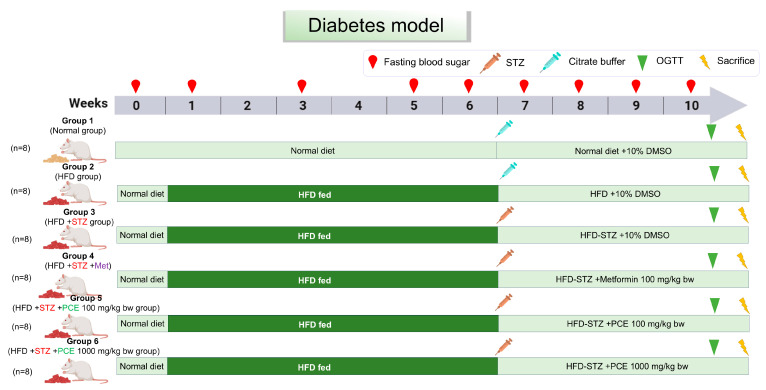
Diabetes simulation and administration of test substances.

**Table 1 ijms-26-08176-t001:** Total amounts of known phenolic and flavonoid compounds in PCE.

Compound	Total Amounts
Rosmarinic acid (mg/g extract)	25.97 ± 0.24
Luteolin (mg/g extract)	9.01 ± 1.19
Apigenin (mg/g extract)	3.25 ± 0.78

Results are presented as mean ± SD of two biological replicates.

**Table 2 ijms-26-08176-t002:** Antioxidant equivalents of PCE.

Assay Methods (Unit of Antioxidant Equivalent)	Antioxidant Capacity
DPPH (mg ascorbic acid equivalent/g of PCE)	94.61 ± 2.87
ABTS (mg Trolox equivalent/g of PCE)	94.91 ± 5.09
FRAP (mg Trolox equivalent/g of PCE)	107.99 ± 9.40
ORAC (mg Trolox equivalent/g of PCE)	668.51 ± 9.85

Results are presented as mean ± SD of three biological replicates.

**Table 3 ijms-26-08176-t003:** Lipid profile in diabetic rats. Results are presented as mean ± SD.

Metabolic Parameter	Total Triglycerides (mg/dL)	Total Cholesterol (mg/dL)	LDL-C (mg/dL)
Normal control	80.25 ± 25.65	66.38 ± 13.16	7.23 ± 1.94
HFD control	66.38 ± 22.67	70.50 ± 8.85	11.57 ± 2.90
HFD-STZ control	193.13 ± 66.21 **	99.63 ± 16.14 ***	14.00 ± 4.47 **
HFD-STZ + Metformin (100 mg/kg bw)	202.88 ± 84.86	99.00 ± 19.13	13.96 ± 2.94
HFD-STZ + PCE 100 mg/kg bw	76.62 ± 27.47 ^aa^	82.12 ± 14.92	14.97 ± 5.22
HFD-STZ + PCE 1000 mg/kg bw	137.88 ± 66.75	99.38 ± 15.52	14.33 ± 2.22

** *p* < 0.01, and *** *p* < 0.001 compared with normal control. ^aa^ *p* < 0.01 compared with HFD-STZ control.

## Data Availability

The data presented in this study are available in the main text and Appendix A.

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
