# Peer review of "Perilla frutescens Seed Residue Extract Restores Gut Microbial Balance and Enhances Insulin Function in High-Fat Diet and Streptozotocin-Induced Diabetic Rats"

_ijms, 2025, doi:10.3390/ijms26178176_

Round 1

Reviewer 1 Report

Comments and Suggestions for Authors

In this manuscript, the perilla seed residue crude extract (PCE) was used to investigate its anti-diabetic effect and gut microbiota composition impact in diabetic rats induced by a high-fat diet (HFD) and streptozotocin (STZ). The topic is interesting and some positive results have been obtained.

To further enhance the clarity of the manuscript and attract readers, some issues need to be improved.

  1. Why does a 100 mg/kg bw dosage exhibit the best anti-diabetic effect rather than 1000mg/kg? Is there a dose-dependent relationship?
  2. Line 90: Are there only compounds rosmarinic acid, luteolin, and apigenin present in the PCE? Or did this manuscript only detect these three? As your statement, the flavonoid content was 67.63 ± 0.67 mg of catechin equivalent/g extract. However, the total of the contents of luteolin and apigenin is about 12 mg/g extract. Some key flavonoids might be missed
  3. Please rewrite the section of Discussion. The relationship between the main components of PCE and the anti-glycation activity (including the growth parameters in diabetic rats, the glucose utilization, the insulin secretion and pathological appearance of the pancreas) as well as the changes in the gut microbiota has not been clearly explained. There are more studies on the anti-glycation effects of flavonoids at the in vitro or cellular level. Therefore, it is necessary to discuss the relationship between the compound content and the positive results of this article from the molecular perspective of the composition of PCE. Additionally, the increasing dosage (from 100 mg/kg bw to 1000 mg) exhibit the downward effect, should also be explained clearly
  4. Figure 2B, the curves of blood glucose levels are overlapping? Is there a significa nt difference in the results?
  5. Section 4.11 which method was used to employ glucose measurement?
  6. In my opinion, Figure 11 should not be placed in the conclusion section, but rather in the discussion section.
  7. Please rewrite the conclusion, the current version is too simplistic

Author Response

We sincerely thank the Editor and all reviewers for their careful evaluation of our manuscript and for providing constructive and insightful comments. We have carefully considered each suggestion and made corresponding revisions to improve the clarity, scientific rigor, and overall quality of the manuscript.

Below, we provide a detailed point-by-point response to each comment. Reviewer comments are shown in italics, followed by our responses in regular text. Changes in the revised manuscript are highlighted for ease of reference.

In this manuscript, the perilla seed residue crude extract (PCE) was used to investigate its anti-diabetic effect and gut microbiota composition impact in diabetic rats induced by a high-fat diet (HFD) and streptozotocin (STZ). The topic is interesting and some positive results have been obtained.

To further enhance the clarity of the manuscript and attract readers, some issues need to be improved.

  1. Why does a 100 mg/kg bw dosage exhibit the best anti-diabetic effect rather than 1000 mg/kg? Is there a dose-dependent relationship?
    • Because we used a crude extract of Perilla seed residue, the exact quantities of all phytochemicals both potential agonists and antagonists are not fully characterized. In this study, the most pronounced anti-diabetic effect was observed at 100 mg/kg body weight. This dose may represent the optimal level at which the phytochemical components of PCE are balanced after metabolism in rats, which could explain why higher doses did not consistently enhance efficacy. Such non-linear dose response patterns are commonly observed with crude extracts in animal models, in contrast to the more predictable relationships seen with pure compounds. Further studies are warranted to precisely identify the active constituents and determine the most effective dosage. So, we describe in discussion on page 14 (Line 378-385).

  1. Line 90: Are there only compounds rosmarinic acid, luteolin, and apigenin present in the PCE? Or did this manuscript only detect these three? As your statement, the flavonoid content was 67.63 ± 0.67 mg of catechin equivalent/g extract. However, the total of the contents of luteolin and apigenin is about 12 mg/g extract. Some key flavonoids might be missed.
    • We agree with the reviewer’s point regarding the likely presence of additional, unidentified compounds in the crude extract of Perilla seed residue. In the present study, we quantified only three known compounds rosmarinic acid, luteolin, and apigenin because authentic standards for these were commercially available. To comprehensively identify other constituents, future analyses using LC-MS/MS are recommended.

According to recent literature, Perilla frutescens seeds may also contain other phenolic acids such as rosmarinic acid-3-O-glucoside, caffeic acid and its derivatives (including novel compounds like 3′-dehydroxyrosmarinic acid), as well as flavonoids such as luteolin-7-O-glucoside and chrysoeriol [1]. For the purposes of this study, we focused on the three compounds with established reference standards and prior reports in Perilla seed.

Reference [1] Yi D, Wang Z, Peng M. Comprehensive Review of Perilla frutescens: Chemical Composition, Pharmacological Mechanisms, and Industrial Applications in Food and Health Products. Foods. 2025;14(7):1252

  1. Please rewrite the section of Discussion. The relationship between the main components of PCE and the anti-glycation activity (including the growth parameters in diabetic rats, the glucose utilization, the insulin secretion and pathological appearance of the pancreas) as well as the changes in the gut microbiota has not been clearly explained. There are more studies on the anti-glycation effects of flavonoids at the in vitro or cellular level. Therefore, it is necessary to discuss the relationship between the compound content and the positive results of this article from the molecular perspective of the composition of PCE. Additionally, the increasing dosage (from 100 mg/kg bw to 1000 mg) exhibit the downward effect, should also be explained clearly.
    • We have revised the discussion to address how each main component of PCE is related to anti-glycation activity, including effects on growth parameters in diabetic rats, glucose utilization, insulin secretion, pancreatic pathology, and changes in gut microbiota (lines 331–343, page 13).

“Rosmarinic acid enhances GLUT4-mediated glucose uptake, inhibits hepatic gluconeogenesis via PEPCK, and reduces proinflammatory cytokines and mediators. In addition to these metabolic effects, it modulates the gut microbiota by increasing beneficial taxa (e.g., Lactobacillus johnsonii, Candidatus Arthromitus, SCFA producers) and reducing potential pathogens such as Escherichia coli and Bifidobacterium pseudolongum in rats. Apigenin improves lipid profiles and suppresses inflammatory markers, and beneficially alters gut microbiota in disease models by increasing beneficial bacteria (e.g., Akkermansia), reducing harmful taxa, and strengthening the intestinal barrier, which protects against inflammation and pathogens. Luteolin also lowers serum triglycerides and LDL-C, suppresses inflammation, and upregulates Irs2 to improve insulin signaling and promotes a healthier gut microbiota and barrier function, reducing systemic inflammation.”

  1. Figure 2B (OGTT), the curves of blood glucose levels are overlapping? Is there a significant difference in the results?
    • In the OGTT curve, each time point reflects a different stage of glucose utilization, beginning with the fasting blood glucose level, which represents the final fasting stage maintained by glycogenolysis and gluconeogenesis. After oral glucose administration, glucose absorption and subsequent uptake occur, leading to changes in blood glucose over time. Therefore, some degree of overlap between time points in different groups can be expected in the OGTT curve. Although the glucose levels at certain individual time points were not significantly different between some groups, the integrated area under the curve (AUC) which represents overall glucose utilization showed statistically significant differences.
  1. Section 4.11 which method was used to employ glucose measurement?
    • Glucometer (Easy G, OK Biotech, Taiwan; strip pad lot S231228D-1) was used to employ glucose measurement. We have already added to the method in line 585-588 on page 19.

  1. In my opinion, Figure 11 should not be placed in the conclusion section, but rather in the discussion section.
    • I agree with your suggestion. Therefore, Figure 11 “A proposed summary of the action of Perilla seed residue on an induced diabetic rat model.” has been already changed to Figure 10 in page 16 after discussion chapter.
  1. Please rewrite the conclusion, the current version is too simplistic
    • The conclusion is rewritten and presented in page 21 line 658 - 667.  

“In conclusion, phytochemicals from perilla seed residue lower fasting blood glucose in diabetic rats by enhancing insulin sensitivity via anti-inflammatory effects, protecting β-cells from the oxidative stress caused by a high-fat diet and streptozotocin, and reducing diabetes-related complications. Notably, these effects were dose-independent, with optimal benefits observed at 100 mg/kg bw. PCE also results in the restoration of beneficial gut bacteria (e.g., Lactobacillus, Clostridium sensu stricto 1, Akkermansia) while reducing pathogenic taxa (e.g., Negativibacillus, Eubacterium nodatum group), indicating improved metabolic health through microbiome rebalancing and reduced inflammation. These findings highlight the potential of perilla-derived products as natural anti-diabetic agents and underscore their value in the field of Thai medicinal herbs.”

Reviewer 2 Report

Comments and Suggestions for Authors

This study investigated the anti-diabetic effect of Perilla frutescens seed residue extract (PCE) on high-fat diet and streptozotocin induced diabetes rats and its effect on intestinal flora. The research design is reasonable, but the therapeutic efficacy of PCE appears to be moderate when compared to conventional treatments. Some of the content needs further clarification and supplementation to enhance the scientific and readability of the paper. 1. The dosages of perilla seed residue crude extract (PCE) are 100 mg/kg and 1000 mg/kg. Why did you increase the dosage so much? The effect of 100 mg/kg and 1000 mg/kg in alleviating diabetes is not very outstanding, so, will the optimal PCE dosage for mitigating diabetes fall within the range of 100-1000 mg/kg. 2. Why choose a dosage of 100 mg/kg for metformin 3. What is the main active substance in PCE? Has PCE been purified? 4. Compared with other extracts, PCE's polyphenol and flavonoid content, as well as its antioxidant activity in vitro, are not very good. Why study PCE? 5. As shown in figure 4D, the intra-group reproducibility of TNF-α is poor, and the anti-inflammatory effect of PCE is limited. Please explain 6. In figure 4E, there are two groups, why not test all groups? 7. In figure 4F, the effect of PCE on the gene expression of inflammatory factors is limited. Please explain. 8. As shown in figure 7B, the composition of the microbial community of normal control and HFD-STZ is similar and not separated. Please explain. 9. In figure 8, the font size of the group is too small to see clearly.

Author Response

We sincerely thank the reviewer for their careful evaluation of our manuscript and for providing constructive and insightful comments. We have carefully considered each suggestion and made corresponding revisions to improve the clarity, scientific rigor, and overall quality of the manuscript.

Below, we provide a detailed point-by-point response to each comment. Reviewer comments are shown in italics, followed by our responses in regular text. Changes in the revised manuscript are highlighted for ease of reference.

This study investigated the anti-diabetic effect of Perilla frutescens seed residue extract (PCE) on high-fat diet and streptozotocin induced diabetes rats and its effect on intestinal flora. The research design is reasonable, but the therapeutic efficacy of PCE appears to be moderate when compared to conventional treatments. Some of the content needs further clarification and supplementation to enhance the scientific and readability of the paper.

  1. The dosages of perilla seed residue crude extract (PCE) are 100 mg/kg and 1000 mg/kg. Why did you increase the dosage so much? The effect of 100 mg/kg and 1000 mg/kg in alleviating diabetes is not very outstanding, so, will the optimal PCE dosage for mitigating diabetes fall within the range of 100-1000 mg/kg.

- Thank you for your question regarding the PCE dosages. The dosages of 100 mg/kg and 1000 mg/kg were selected based on the amounts of rosmarinic acid and other active compounds present in the extract, with rosmarinic acid being a key component of perilla seed residue. Previous studies, including Den Hartogh et al. (2023) [1], have shown that rosmarinic acid at range 1.80 mg/mL can reduce blood glucose and enhance glucose uptake in cells with similar levels found in 100 and 1,000 mg/Kg of PCE (0.75 mg/kg and 7.50 mg/kg in low and high dose, respectively). Therefore, we aimed to evaluate whether different concentrations of PCE, corresponding to varying amounts of rosmarinic acid, would affect diabetes outcomes.

Refererence [1] Den Hartogh DJ, Vlavcheski F, Tsiani E. Muscle Cell Insulin Resistance Is Attenuated by Rosmarinic Acid: Elucidating the Mechanisms Involved. Int J Mol Sci. 2023;24(6)

  1. Why choose a dosage of 100 mg/kg for metformin

-A dose of 100 mg/kg was selected because it is effective, well-tolerated, and widely supported by previous studies in diabetes research using the HFD-STZ rat model. In addition, this dosage follows the our previous studies done by Sakuludomkan W. et al. (2020) [2] which show the effective concentration for anti-diabetes in this model.

References [2] Sakuludomkan W, Yeewa R, Subhawa S, Khanaree C, Bonness AI, Chewonarin T. Effects of Fermented Houttuynia cordata Thunb. on Diabetic Rats Induced by a High-Fat Diet with Streptozotocin and on Insulin Resistance in 3T3-L1 Adipocytes. J Nutr Metab. 2021;2021:6936025.

  1. What is the main active substance in PCE? Has PCE been purified?

- The known active compounds identified in PCE are rosmarinic acid, apigenin, and luteolin, with their concentrations presented in Table 1 (page 3, line 101). The major compound is rosmarinic acid, present at the highest concentration (25.97 mg/g extract). As our aim was to utilize all bioactive components in perilla seed residue for potential development as a food supplement, we employed a crude extract in this study rather than isolating and testing single purified compounds. However, the use of crude extract from periila seed residue is added to the objective of study on page 2 line 84.

  1. Compared with other extracts, PCE's polyphenol and flavonoid content, as well as its antioxidant activity in vitro, are not very good. Why study PCE?

- Our rationale for studying PCE stems from our aim to add value to Perilla seed residue, a byproduct of food production that is typically discarded. PCE contains several phytochemicals with reported biological activities relevant to diabetes prevention and management, as described in the Introduction (page 2, lines 76-81) and Discussion (page 13, lines 333-345). Moreover, the crude extract likely contains numerous unidentified compounds that may exert synergistic or antagonistic effects in vivo. For this reason, direct comparison with extracts from other herbs or natural products can be challenging, as the research questions and intended biological targets often differ.

While the polyphenol and flavonoid content, as well as in vitro antioxidant activity, of PCE may appear modest compared with certain other extracts, these assays were used primarily as screening tools to confirm the presence and potential activity of bioactive compounds. These compounds may exert anti-diabetic effects in vivo through mechanisms beyond direct antioxidant activity particularly anti-inflammatory pathways and the stimulation of glucose uptake, both of which are critical for improving insulin sensitivity.

In our previous work, PCE demonstrated anti-inflammatory activity in an inflammation-induced rat colon cancer model by inhibiting macrophage activation and suppressing the inflammatory response in colonic epithelial cells [3]. In addition, our preliminary, unpublished data indicate that PCE reduces inflammation in activated macrophage cell lines and enhances glucose uptake in inflammation-induced insulin-resistant adipocytes at concentrations of 100 and 200 μg/mL. Based on these findings, we selected PCE for evaluation in an anti-diabetic model. 

Reference [3] Chantana W, Hu R, Buddhasiri S, Thiennimitr P, Tantipaiboonwong P, Chewonarin T. The Extract of Perilla frutescens Seed Residue Attenuated the Progression of Aberrant Crypt Foci in Rat Colon by Reducing Inflammatory Processes and Altered Gut Microbiota. Foods. 2023;12(5):988.

  1. As shown in figure 4D, the intra-group reproducibility of TNF-α is poor, and the anti-inflammatory effect of PCE is limited. Please explain

- In animal models, variability between individual subjects is common, particularly when using ELISA to measure cytokines, due to the method’s sensitivity limitations. In our study, one to two rats exhibited outlier values for TNF-α, which increased the standard deviation (SD) within the group and may have reduced the statistical significance of PCE’s effect.

Additionally, it should be noted that systemic inflammation rises rapidly after STZ injection. Serum inflammatory cytokine levels measured by ELISA at the time of sacrifice reflect the overall inflammatory status at that terminal stage, rather than the peak inflammatory response. As a result, the high inflammatory state in the positive control group may not have been fully captured, which could contribute to the apparent limitation in the observed anti-inflammatory effect of PCE.

  1. 6. In figure 4E, there are two groups, why not test all groups?

- The purpose of including Figure 4E was to illustrate the effect of a high-fat diet (HFD) in inducing inflammation in adipose tissue, which contributes to the development of insulin resistance prior to streptozotocin (STZ) administration thereby mimicking type 2 diabetes. Inflammation in adipose tissue was not pronounced in this model, which may explain why the anti-inflammatory effect of PCE was not clearly observed in this parameter. Consequently, we did not present PCE-treated groups in this figure.

However, because the gut microbiome data required comparison between the HFD and HFD-STZ groups, Figure 4E was included to provide relevant context for the inflammation status in these groups.

  1. In figure 4F, the effect of PCE on the gene expression of inflammatory factors is limited. Please explain.

- In Figure 4F, inflammation was assessed in pancreatic tissue to evaluate the effect of STZ on inflammatory gene expression. STZ is a toxic glucose analog that selectively targets pancreatic β-cells via the GLUT2 transporter, inducing DNA damage and apoptosis. This tissue injury upregulates pro-inflammatory enzymes such as iNOS and COX-2, which produce mediators including nitric oxide (NO) and prostaglandins that contribute directly to cellular damage and inflammation.

Given the timing of measurement, the peak effect of PCE on inflammation may have occurred within 1-2 weeks after STZ injection. However, pancreatic tissue was collected at the sacrifice time point, which may have missed this peak, making the anti-inflammatory effect less apparent. Furthermore, fold changes in mRNA expression reflect relative rather than absolute transcript copies numbers, and experimental variation was relatively high in the HFD-STZ group. As a result, statistical significance was not observed. Nonetheless, a residual anti-inflammatory trend was noted in the PCE-treated group, which appeared comparable to the effect of metformin.

  1. As shown in figure 7B, the composition of the microbial community of normal control and HFD-STZ is similar and not separated. Please explain.

- The PCA plot provides a visual projection of the separation among groups based on microbial community composition. When all three groups are analyzed together, the clustering between the normal control and HFD-STZ groups appears less distinct due to overlap with the third group. However, when comparing only the normal control and HFD-STZ groups, a clearer separation is observed, as shown in Supplementary Figure S3.

  1. In figure 8, the font size of the group is too small to see clearly.

-We have already adjusted scale in Figure 8 to see clearly. 

Reviewer 3 Report

Comments and Suggestions for Authors

Summary: This study investigated the potential health benefits of perilla seed residue crude extract (PCE) in a diabetic rat model. The authors showed that PCE improved glucose metabolism, reduced triglyceride levels, improved pancreatic histology and inflammation, and altered gut microbiota composition, suggesting a possible prebiotic effect. Overall, this is a well-designed study with extensive data. That said, I recommend major revisions to improve the clarity of data presentation and to better support some of the key conclusions.

Comments:

  1. I recommend focusing the statistical comparisons and main figures on the HFD+STZ groups (i.e., HFD+STZ alone, +Met, +PCE 100, +PCE 1000), since that’s the core of the study. Including the normal control and HFD-only groups in the main plots makes the comparison harder to interpret and shifts attention away from the therapeutic effects of PCE. These control groups can still be shown in the supplementary data. If the authors agree, this would require some restructuring of the figures and text.
  2. Line 542 mentions PCE was given at 100 and 1,000 mg/kg bw for four weeks — but how often was it administered? Once per day?
  3. Please list the actual amount or percentage of each ingredient used in the high fat diet in Table S2.
  4. Please annotate the purple bar shown in Figure 4 in the legend.
  5. For Figures 5 – 7, please consider updating the box plots to make the presentation of figures more cohesive throughout the manuscript. Also, please consider increasing font sizes for axes and p-values for better readability. Also, make sure the alpha diversity plots are properly cited in the text.
  6. The color coding in the box plots vs. beta diversity plots (Figures 5–7) is inconsistent. Please try to align the color scheme across figures for a more cohesive presentation.
  7. The HFD+STZ+Met group seems to be missing from the microbiome-related figures. If these data are available, it would be helpful to include them. If not, please provide justifications for its exclusion.
  8. Please consider adding HPLC chromatograms of the known active compounds in PCE to the supplementary. Also, was an internal standard used for quantification? In Section 4.6, it's mentioned that assays were run in duplicate with three independent tests — does this also apply to the other assays described in Sections 4.2–4.5? This could be clarified, especially since SDs are reported in Table 2.
  9. Since both the gut microbiota and metabolic outcomes were influenced by PCE, I think it would make sense to conduct a correlation analysis between key diabetes markers and the microbial taxa that were enriched. This would help strengthen the link between microbiome shifts and host metabolic improvements.
  10. The manuscript mentions that the total cholesterol wasn’t much affected by PCE. If possible, I suggest measuring HDL-C and LDL-C to give a fuller picture of lipid outcomes, especially considering the proposed prebiotic effect.
  11. Lines 373–375 and 413–415 mention that the low dose of PCE (100 mg/kg) had more pronounced effects than the higher dose. That’s quite interesting, especially since the two doses showed noticeable differences in microbiome and host metabolism. I’d suggest expanding on this point a bit more. What are the potential compounds in the extract that could have inhibitory effects at higher doses?
  12. Since the authors showed enrichment of SCFA-producing taxa in PCE groups, I wonder if SCFA levels were measured. If so, it would be helpful to include them. If not, this could be worth considering, as it would support the idea of a functional prebiotic effect.

Author Response

We sincerely thank the Editor and all reviewers for their careful evaluation of our manuscript and for providing constructive and insightful comments. We have carefully considered each suggestion and made corresponding revisions to improve the clarity, scientific rigor, and overall quality of the manuscript.

Below, we provide a detailed point-by-point response to each comment. Reviewer comments are shown in italics, followed by our responses in regular text. Changes in the revised manuscript are highlighted for ease of reference.

Summary: This study investigated the potential health benefits of perilla seed residue crude extract (PCE) in a diabetic rat model. The authors showed that PCE improved glucose metabolism, reduced triglyceride levels, improved pancreatic histology and inflammation, and altered gut microbiota composition, suggesting a possible prebiotic effect. Overall, this is a well-designed study with extensive data. That said, I recommend major revisions to improve the clarity of data presentation and to better support some of the key conclusions.

Comments:

  1. I recommend focusing the statistical comparisons and main figures on the HFD+STZ groups (i.e., HFD+STZ alone, +Met, +PCE 100, +PCE 1000), since that’s the core of the study. Including the normal control and HFD-only groups in the main plots makes the comparison harder to interpret and shifts attention away from the therapeutic effects of PCE. These control groups can still be shown in the supplementary data. If the authors agree, this would require some restructuring of the figures and text.
    • Thank you very much for your thoughtful suggestion. We agree that the primary objective of our study is to evaluate the effects of PCE in HFD+STZ-induced diabetic rats, and that statistical comparisons among the HFD+STZ groups are central to our investigation. However, we believe that including the normal control and HFD-only groups in the main figures is essential for providing a comprehensive understanding of disease progression in our model. Specifically, comparison with the normal control group is necessary to validate the successful induction of diabetes in our HFD+STZ group, while inclusion of the HFD-only group allows us to confirm the development of insulin resistance prior to diabetes onset. These comparisons support our discussion of the underlying mechanisms of PCE and reinforce the validity of our animal model. That said, we appreciate your perspective and remain open to restructuring the figures and text if the editorial team deems it more appropriate. We are also willing to present these control groups in the supplementary material as suggested, should this be preferred.

  1. Line 542 mentions PCE was given at 100 and 1,000 mg/kg bw for four weeks — but how often was it administered? Once per day?
    • Thank you for your question. PCE was prepared in two concentrations corresponding to doses of 100 and 1,000 mg/kg body weight, in order to standardize the volume administered. These PCE suspensions were given to the rats once daily in the morning for four consecutive weeks. The details of the dosing regimen are provided on page 19, line 566-567 of the manuscript.

  1. Please list the actual amount or percentage of each ingredient used in the high fat diet in Table S2.
    • Thank you for your suggestion. We have now included the actual amounts and percentages of each ingredient used in the high-fat diet in Table S2, as requested.

  1. Please annotate the purple bar shown in Figure 4 in the legend.
    • As requested, the purple bar in Figure 4 has now been explained in the figure legend.

  1. For Figures 5 – 7, please consider updating the box plots to make the presentation of figures more cohesive throughout the manuscript. Also, please consider increasing font sizes for axes and p-values for better readability. Also, make sure the alpha diversity plots are properly cited in the text.
    • Thank you for your valuable suggestions. Figures 5-7 have been revised to enhance consistency and cohesion in the presentation of the box plots throughout the manuscript. Font sizes for axes and p-values have also been increased to improve readability. Additionally, the alpha diversity data are now properly cited in the text on page 9, line 258-279.

  1. The color coding in the box plots vs. beta diversity plots (Figures 5–7) is inconsistent. Please try to align the color scheme across figures for a more cohesive presentation.
    • Thank you for your valuable suggestion. We have adjusted the color schemes in the box plots to ensure consistency with the assigned groups across Figures 5-7 (pages 9-11). This should provide a more cohesive and clear presentation of the data.

  1. The HFD+STZ+Met group seems to be missing from the microbiome-related figures. If these data are available, it would be helpful to include them. If not, please provide justifications for its exclusion.
    • Thank you for raising this point. We acknowledge the growing interest in the gut microbiome as a mediator of metformin’s therapeutic effects, with evidence from both clinical and preclinical studies demonstrating metformin’s impact on gut microbial composition and metabolic outcomes [4]. However, in this study, metformin was included as a positive control specifically for its well-established anti-diabetic effects rather than for comparative analysis of gut microbiome alterations. Therefore, we did not perform microbiome analyses in the HFD+STZ+Met group. We focused our microbiome investigations on groups relevant to evaluating the effects of PCE. We appreciate your understanding and are happy to clarify this rationale in the revised manuscript.

Reference [4] Zhou X, Zhou J, Ban Q, Zhang M, Ban B. Effects of metformin on the glucose regulation, lipid levels and gut microbiota in high-fat diet with streptozotocin induced type 2 diabetes mellitus rats. Endocrine. 2024;86(1):163-72.

  1. Please consider adding HPLC chromatograms of the known active compounds in PCE to the supplementary. Also, was an internal standard used for quantification? In Section 4.6, it's mentioned that assays were run in duplicate with three independent tests — does this also apply to the other assays described in Sections 4.2–4.5? This could be clarified, especially since SDs are reported in Table 2.
    • Thank you for your valuable suggestions. The HPLC chromatograms of the known active compounds in PCE have been included in the supplementary data as requested. An internal standard was not used for quantification; instead, established standards were employed, allowing us to compare retention times and peak areas in accordance with standard HPLC protocols. Additionally, the assays described in Sections 4.2-4.5 were also performed in duplicate with three independent tests to generate mean ± SD values. This clarification has now been added to Sections 4.2-4.5 of the revised manuscript.

  1. Since both the gut microbiota and metabolic outcomes were influenced by PCE, I think it would make sense to conduct a correlation analysis between key diabetes markers and the microbial taxa that were enriched. This would help strengthen the link between microbiome shifts and host metabolic improvements.
    • Thank you for your suggestion. We agree with your point and have conducted a correlation analysis between key diabetes markers and the enriched microbial taxa. The results are presented as a heatmap in Supplementary Figure S4, titled “Heatmap correlation between key metabolic markers and gut microbiome.” These correlations are described in the main text on page 12, lines 312-318, with a corresponding citation to Supplementary Figure S4.

  1. The manuscript mentions that the total cholesterol wasn’t much affected by PCE. If possible, I suggest measuring HDL-C and LDL-C to give a fuller picture of lipid outcomes, especially considering the proposed prebiotic effect.
    • Thank you for your suggestion. In diabetes, the depletion of insulin leads to defective lipid metabolism, particularly affecting triglyceride synthesis and its transport to peripheral tissues via VLDL. Therefore, the serum lipid parameter most relevant to our hypothesis is total triglyceride, which is closely related to VLDL-cholesterol. To address your suggestion, LDL-cholesterol an intermediate lipoprotein formed after the hydrolysis of fatty acids and glycerol from VLDL has been added to Table 3 and described in the text on page 8, lines 231–240. These data confirm that PCE affects triglyceride levels in VLDL in relation to insulin action but does not influence cholesterol transport in serum via LDL. HDL was not measured in rat serum due to methodological limitations. However, HDL, which reflects reverse cholesterol transport, could be estimated indirectly by calculation, and we expect that it would not be significantly altered by PCE administration in this diabetic rat model.

  1. Lines 373–375 and 413–415 mention that the low dose of PCE (100 mg/kg) had more pronounced effects than the higher dose. That’s quite interesting, especially since the two doses showed noticeable differences in microbiome and host metabolism. I’d suggest expanding on this point a bit more. What are the potential compounds in the extract that could have inhibitory effects at higher doses?
    • Because we used a crude extract of Perilla seed residue, the exact quantities of all phytochemicals both potential agonists and antagonists are not fully characterized. In this study, the most pronounced anti-diabetic effect was observed at 100 mg/kg body weight. This dose may represent the optimal level at which the phytochemical components of PCE are balanced after metabolism in rats, which could explain why higher doses did not consistently enhance efficacy. Such non-linear dose-response patterns are commonly observed with crude extracts in animal models, in contrast to the more predictable relationships seen with pure compounds. Further studies are warranted to precisely identify the active constituents and determine the most effective dosage. So, we describe in discussion on page 14 (Line 380-387)

  1. Since the authors showed enrichment of SCFA-producing taxa in PCE groups, I wonder if SCFA levels were measured. If so, it would be helpful to include them. If not, this could be worth considering, as it would support the idea of a functional prebiotic effect.
    • Thank you for raising this important point. One of our aims was indeed to determine whether alterations in the gut microbiota could influence host metabolism. However, due to time and funding constraints, we did not measure SCFA levels directly in this study. We agree that quantifying SCFAs would provide valuable supporting evidence for the functional prebiotic effect of PCE, particularly in light of the observed enrichment of SCFA-producing taxa. We will consider including direct SCFA measurements in future experiments to further strengthen our findings.

Round 2

Reviewer 1 Report

Comments and Suggestions for Authors

Now I think it is suitable for publish in IJMS

Reviewer 2 Report

Comments and Suggestions for Authors

ok

Reviewer 3 Report

Comments and Suggestions for Authors

Most of my previous comments and suggestions have been appropriately addressed.